# COVID-19 Infection-Related Weight Loss Decreases Eating/Swallowing Function in Schizophrenic Patients

**DOI:** 10.3390/nu13041113

**Published:** 2021-03-29

**Authors:** Takeshi Kikutani, Yoko Ichikawa, Eri Kitazume, Arato Mizukoshi, Takashi Tohara, Noriaki Takahashi, Fumiyo Tamura, Manami Matsutani, Junko Onishi, Eiichiro Makino

**Affiliations:** 1Division of Clinical Oral Rehabilitation, Graduate School of Life Dentistry, The Nippon Dental University, Tokyo 184-0011, Japan; 2Tama Oral Rehabilitation Clinic, The Nippon Dental University School of Life Dentistry, Tokyo 184-0011, Japan; psalm236yi@tky.ndu.ac.jp (Y.I.); mizukoshi@tky.ndu.ac.jp (A.M.); ndu_061352@tky.ndu.ac.jp (T.T.); high-bri@tky.ndu.ac.jp (N.T.); fumita@tky.ndu.ac.jp (F.T.); 3Department of Oral and Maxillofacial Surgery, The Nippon Dental University Hospital, Tokyo 102-8158, Japan; kitazume@tky.ndu.ac.jp; 4Musashino Central Hospital, Tokyo 184-8585, Japan; rin.03.ee@gmail.com (M.M.); onishi_m_c_h@cap.ocn.ne.jp (J.O.); mch-cond@ah.wakwak.com (E.M.)

**Keywords:** COVID-19, dysphagia, cachexia, weight loss, schizophrenia

## Abstract

Background: In older people with psychoneurological diseases, COVID-19 infection may be associated with a risk of developing or exacerbating dysphagia. The aim of the present study was to examine the relationship between eating/swallowing function and COVID-19 infection. Methods: Subjects were 44 inpatients with confirmed COVID-19 infection being treated for schizophrenia in a psychiatric ward. Eating function was assessed using the Food Intake Level Scale (FILS) before and after infection. We also evaluated age, comorbidities, COVID-19 hospital stay, obesity index, weight loss rate, and chlorpromazine equivalent. Results: Subjects had a mean age of 68.86 years. Pre-infection, 20 subjects had a FILS score of 7–9 (presence of eating/swallowing disorder) and 24 subjects had a score of 10 (normal). Eating function after infection resolution showed decreasing FILS score compared to that before infection in 14 subjects (74.14 years). Six subjects (79.3 years) transitioned from oral feeding to parenteral feeding. A ≥ 10% weight loss during infection treatment was significantly associated with decreased eating function and a transition to parenteral feeding. Chlorpromazine equivalents, comorbidities, and number of days of hospitalization showed no associations with decreased eating function. Conclusions: Preventing malnutrition during treatment for COVID-19 infection is important for improving post-infection life prognosis and maintaining quality of life (QOL).

## 1. Introduction

The COVID-19 pandemic has caused more than 70 million infections and more than 1.6 million deaths as of December 2020. Globally, around 20 million people are thought to be suffering from schizophrenia and receiving pharmacotherapy and/or psychosocial intervention [1]. The prevalence of COVID-19 is known to be high among patients with psychiatric diseases such as schizophrenia [2], with this high prevalence attributed to difficulties in various tasks and behaviors that are important for reducing infection risks, such as hand washing, social distancing, and quarantine [3,4]. Moreover, the prognosis of COVID-19 patients is known to be worse among individuals with psychiatric diseases than among those without such diseases [5]. Dysphagia appears commonly in schizophrenic patients [6].

SARS-CoV-2, the causative pathogen for COVID-19, is known to be neurotropic and neuroinvasive [7] and is thus likely to have effects on swallowing function. Furthermore, various psychiatric symptoms have been reported in patients during hospital admission for treatment of COVID-19 infection while receiving treatment with medications including antipsychotics [8], which may also greatly affect swallowing function. COVID-19 is known to cause respiratory symptoms as well as gastrointestinal symptoms, which can lead to malnutrition [9].

It has been pointed out that COVID-19 leads to muscle wastage through a highly catabolic state [10]. Loss of muscle mass and strength (i.e., sarcopenia) is a systemic process that affects not only the lower extremities but also the swallowing muscles [11].

There are few reports of cases showing such a relationship [12]. The present study examined relationships between eating/swallowing function and various factors in a population of patients who developed COVID-19 infection in a psychiatric ward, to shed light on the effects of COVID-19 infection on swallowing function.

## 2. Materials and Methods

### 2.1. Study Design and Setting

This was a retrospective observational study of patients with confirmed COVID-19 who were admitted to the psychiatric ward of a hospital located in the suburbs of Tokyo, Japan in May or June of 2020.

### 2.2. Participants

All subjects were undergoing inpatient treatment due to schizophrenia. Among the subjects with confirmed infection during the observation period, 44 subjects were included in the present study, after excluding one subject receiving nutrition via parenteral feeding before infection, four subjects who died in another hospital after being transferred for treatment, and one patient who was transferred to another hospital after resolution of COVID-19.

### 2.3. Variables

#### 2.3.1. Basic Information

The age and comorbidities of the subjects before infection were investigated. Subjects were assigned separate according to age, ≥70 years and <70 years. Comorbidities were described and characterized based on the Charlson Comorbidity Index (CCI), with stage 0 indicating low grade; stage 1–2, medium; stage 3–4, high; stage ≥5, very high [13]. The activities of daily living (ADL) of the subjects before and after infection were investigated. The ADL determined whether subjects required assistance with walking and eating. Descriptive information included the number of days in the hospital to which the patient was transferred for COVID-19 treatment, whether oxygen was administered, and whether endotracheal intubation was performed. Hospital stays for treatment of infections longer than 30 days were considered long-term hospitalizations, while those less than 30 days were considered short-term.

#### 2.3.2. Status of Pharmacotherapy

Medications used before infection and during treatment for COVID-19 were described, and the use of ≥6 medicines was defined as polypharmacy [14]. The chlorpromazine-equivalent value (CPZ-equivalent) per patient was used to calculate the daily antipsychotic dose per patient based on chlorpromazine dose equivalents for antipsychotics. Antipsychotic doses of ≥600 or ≥1000 mg CPZ-equivalents were considered high doses [15].

#### 2.3.3. Nutritional Status

Body mass index (BMI) and weight loss at the time of infection were determined based on height and weight at the month of infection and weight six months before infection. Subjects ≥70 years old with a BMI <17.0 kg/m^2^ and those <70 years old with a BMI <17.8 kg/m^2^ were considered malnourished [16]. Subjects with a rate of weight loss from six months before infection of ≥5% or ≥10% were also considered malnourished [17]. Similarly, BMI at the time of infection cure and the rate of weight loss were determined based on those during infection to identify malnutrition based on the same criteria.

#### 2.3.4. Eating Function

The eating function of subjects was evaluated based on the Food Intake Level Scale (FILS) [18]. The FILS, a 10-point observer-rating scale for assessing dysphagia, was used to classify the severity of dysphagia as follows: score: 1–3, “no oral intake”; score: 4–6, “oral intake and alternative nutrition”; score: 7–9, “oral intake alone”; score 10, “normal oral food intake”. Subjects with a change in FILS score from 10 pre-infection to 1–9 and those with a change in score from 7–9 pre-infection to 1–6 were considered to have decreased eating function. Moreover, those who showed a change to a score of 1–3 were transitioned to parenteral feeding.

#### 2.3.5. Pharyngeal Movements and Dysosmia/Dysgeusia after Infection Cure

Two months after infection cure, subjects who transitioned to parenteral feeding underwent a videoendoscopic swallowing examination to check for the presence of velopharyngeal insufficiency during swallowing. In addition, subjects were interviewed about potential olfactory and gustatory dysfunctions.

### 2.4. Statistical Analysis

Changes in eating function from before infection were analyzed by the Kruskal–Wallis test. Chi-square tests were performed to examine the relationships between each factor and subjects with decreased eating function and conversion to parenteral feeding, respectively. A hazard ratio of <5% was considered to denote significance.

## 3. Results

### 3.1. Participant Characteristics

The 44 subjects (mean age, 68.86 ± 12.74 years) comprised 13 men (mean age, 72.1 ± 10.3 years; range, 59–93 years) and 31 women (mean age, 67.5 ± 13.6 years; range, 30–89 years). Mean CCI was 0.73 ± 1.20, categorized as low in 25 subjects, medium in 17, high in one, and very high in one. One subject received assistance with eating, and seven had difficulty walking independently.

#### 3.1.1. Medication Use Status and Nutritional Status before Infection

Before infection, the mean number of drugs used was 7.52 ± 3.77, and 31 subjects were defined as receiving polypharmacy. Moreover, the mean antipsychotic dose in terms of CPZ-equivalents was 454.94 ± 455.87 mg, with ≥600 and ≥1000 mg CPZ-equivalents in 14 and six subjects, respectively. Eleven subjects were diagnosed as malnourished on the basis of BMI, and three subjects by weight loss rate.

#### 3.1.2. Relationships between Eating Status and Age, Nutrition, and Pharmacotherapy before Infection

Before infection, the FILS score was 7 in five subjects, 8 in 11 subjects, 9 in four subjects, and 10 in 24 subjects, indicating that 20 subjects had an eating/swallowing disorder, and 24 had normal eating/swallowing function. Eating function was significantly associated with age, and showed no relationship to weight loss, BMI-based nutritional status, or CPZ-equivalent dose (Table 1). 

### 3.2. Relationship between Eating Status and Various Factors after Infection Care

#### 3.2.1. Treatment for Infection

The mean duration of hospitalization for COVID-19 infection was 32.1 ± 19.1 days (range: 8–83 days). During treatment for infection, three subjects received endotracheal intubation, and 15 received oxygen administration.

#### 3.2.2. ADL and Drug Use Status after Infection Care

After infection cure, the number of subjects requiring assistance with eating had increased significantly to seven (*p* = 0.014), and the number of subjects experiencing difficulty with walking independently had increased significantly to 18 (*p* < 0.01). Twenty-one subjects were considered to be receiving polypharmacy. Furthermore, the antipsychotic dose was ≥600 mg CPZ-equivalents in 19 subjects and ≥1000 mg CPZ-equivalents in five subjects.

#### 3.2.3. Nutritional Status after Recovery from Infection

After full recovery from COVID-19 infection, 13 subjects were diagnosed as malnourished on the basis of BMI, and 11 subjects by weight loss rate. The number of malnourished subjects who were diagnosed on the basis of weight loss rate had increased significantly compared to that before COVID-19 infection.

#### 3.2.4. Change in Eating Function before and after Recovery from Infection

After infection care, FILS score was 2 in six subjects, 7 in five subjects, 8 in 16 subjects, 9 in two subjects, and 10 in 15 subjects, showing significant decreases in eating function from the pre-infection assessment (*p* < 0.01). Eating function after infection care had decreased from that before infection in 14 subjects (mean age, 74.14 ± 9.91 years). Parenteral nutrition was initiated in six subjects (mean age, 79.3 ± 10.71 years), and a videoendoscopic swallowing examination revealed that four of them developed dysphagia during swallowing. Among the 24 subjects with normal function (FILS score, 10) before infection, one transitioned to parenteral feeding (score, 1–3) and eight were still able to orally ingest food, but had developed an eating/swallowing disorder (score, 7–9). Among the 20 subjects who were able to eat orally but had an eating/swallowing disorder (FILS score, 7–9) before infection, five had transitioned to parenteral feeding (score, 1–3) (Table 2). Of the six subjects who transitioned to parenteral feeding, two complained of olfactory and gustatory dysfunctions. Four subjects were unavailable for interviews to determine whether they had such symptoms, due to decreased cognitive function.

#### 3.2.5. Relationship between Changes in Eating Status and Various Factors

Weight loss of ≥10% after infection care of COVID-19 showed an association with both worsening of eating status compared with the pre-infection assessment (*p* < 0.01) and conversion to parenteral feeding (*p* < 0.05) (Table 3).

#### 3.2.6. Clinical Findings of Transition to Parenteral Feeding

Fiberoptic endoscopic evaluation of swallowing confirmed disorders of pharyngeal movement in three of the six subjects in the present study who transitioned to parenteral feeding, while two of the subjects who transitioned to tube feeding complained of dysgeusia.

## 4. Discussion

In the present study, the effects of COVID-19 infection on eating/swallowing function were clarified through the eating/swallowing disorders observed in COVID-19-infected subjects in a psychiatric ward. The COVID-19 infection significantly led to decreased eating function. Furthermore, 13.6% of subjects transitioned to enteral nutrition. Subjects who already had eating/swallowing disorders before infection experienced marked decreases in swallowing function. During hospital admission, 25% of subjects experienced marked weight loss. COVID-19-related weight losses were significantly associated with both a decrease in eating function and difficulty with oral ingestion. In the present study, before COVID-19 infection, the dysphagia severity scale in a psychiatric cohort showed a relationship of age to severity, not to nutritional status or the use of antipsychotics. Others have found that the use of antipsychotics to treat schizophrenia was correlated with dysphagia [6]. Haga et al. [19] examined the factors associated with the onset of pneumonia in schizophrenic patients in Japan. They found that, in addition to high doses of antipsychotics in terms of CPZ equivalents, malnutrition and age also showed relationships. One explanation might be that appropriate dosages for psychiatric stabilization had been achieved, and that few subjects were morbidly undernourished beforehand.

In the present study subjects, COVID-19 significantly affected eating function. SARS-CoV-2, the causative pathogen for COVID-19, is known to be neurotropic and neuroinvasive [7]. The virus attacks nerves directly, resulting in olfactory and gustatory dysfunctions as part of the effect on the glossopharyngeal and vagus nerves [20]. In addition, disorders of pharyngeal sensation and movement have also been reported as an effect on the glossopharyngeal and vagus nerves [21]. Complaints of olfactory and gustatory dysfunctions could not be confirmed in many subjects in the present study, potentially because of unavailability for interviews due to schizophrenia-related communication issues. However, the present study suggests that dysgeusia may be related to dysphagia. In addition, a fiberoptic endoscopy revealed pharyngeal-based disorders. Furthermore, among some subjects with severe COVID-19 receiving treatments such as endotracheal intubation, the presence of so-called “postintubation dysphagia” after COVID-19 infection resolution has also been discussed [22]. While some subjects in the present study received endotracheal intubation due to COVID-related pneumonia, the association with subsequent dysphagia remains unclear.

During treatment for COVID-19, 43% of subjects in the present study experienced ≥5% weight loss, and 25% of subjects showed ≥10% weight loss. The subject of this study also suffered from the risk of malnutrition by COVID-19 infection. Subjects who had a ≥10% weight loss showed a significant association with decreased eating function and transition to parenteral feeding. Allard et al. [23] reported that a large number of patients admitted due to COVID-19 had malnutrition. COVID-19 is also known to cause gastrointestinal symptoms, diarrhea, mild abdominal pain, nausea, vomiting, and loss of appetite, making it a potential cause of nutritional disorders [24]. Moreover, the systemic inflammatory response of cytokine storm is directly related to increased muscle protein breakdown, albumin consumption, and a metabolic disorder of major nutrients, which may contribute to malnutrition and the onset of cachexia [25,26]. These reports indicate that malnutrition is a risk for COVID infection, and that the infection increases the risk of malnutrition. A metabolic syndrome characterized by loss of muscle in association with an underlying illness is known as cachexia. A major clinical feature of cachexia in adults is weight loss, which has been linked to inflammation and loss of appetite [27]. COVID-19-induced gastrointestinal symptoms, anorexia, and cytokine storm are known to cause weight loss and to trigger cachexia [28]. While no measurements of skeletal muscle mass were conducted in the present study, weight loss during treatment for COVID-19 can be attributed to cachexia, which is believed to lead to a decrease in the strength of swallowing-related muscles and to have effects on swallowing function [29]. Malnutrition is associated with decreased eating/swallowing function and can lead to dysphagia, which is one cause of malnutrition, in turn contributing to further malnutrition [30]. Preventing malnutrition during treatment for COVID-19 infection is important to maintain eating/swallowing function.

This study has several limitations. First, the sample, consisting of COVID-19 cases that arose in a single hospital ward, was too small to allow for adequate statistical analyses. In addition to significant weight loss, other factor such as the subject’s age, gender, COVID-19 severity, Chlorpromazine-equivalent may affect feeding function. However, in this study, these could not be removed as confounding factors. In order to generalize the results of this manuscript, it is necessary to carry out research using a large number of subjects. Nevertheless, the present study derived its strength from the fact that subjects were inpatients in the same hospital ward and received the same quality of medical care before the onset of the infection. Second, the study was based on pre-infection clinical records in the psychiatric ward and infection care assessments of eating function, as no detailed data were available on the treatment and nutritional management provided during the treatment of COVID-19 infection. These data, although important in terms of the prophylaxis of the COVID-19 infection-induced cachexia, could not be used in the present investigation.

## 5. Conclusions

In schizophrenic patients, who are already prone to experiencing decreased swallowing function, weight loss due to COVID-19 infection is a major risk factor for further decreases in eating/swallowing function. Preventing malnutrition during treatment for COVID-19 infection is important to improve post-infection prognosis and maintain QOL.

## Figures and Tables

**Table 1 nutrients-13-01113-t001:** Relationships between eating status and age, nutrition, and pharmacotherapy before infection.

		FILS				
Characteristics	Overall	7 (*n* = 5)	8 (*n* = 11)	9 (*n* = 4)	10 (*n* = 24)	*p*-Value
Age	71.0 (60.0–78.8)	84.0 (80.5–91.0)	75.0 (65.0–80.0)	77.5 (74.8–78.8)	62.5 (56.3–71.0)	<0.001
Weight loss (%)	−0.8 (−0.3–2.1)	0.17 (−4.1–7.2)	−2.6 (−4.3–0.2)	1.5 (−9.5–1.5)	−0.2 (−2.7–2.7)	0.77
BMI	20.8 (17.5–23.8)	17.3 (15.1–22.8)	18.4 (15.9–20.7)	21.1 (16.2–24.5)	22.5 (18.8–24.2)	0.137
CPZ equivalent	305 (100–737.5)	140.0 (50.0–12.5)	300.0 (37.9–800.0)	150.0 (25.0–275.0)	402.5 (131.3–875.0)	0.123
Number of medications	7.5 (5.0–11.0)	8.0 (5.0–12.5)	9.0 (7.0–11.0)	6.0 (3.25–8.75)	6.5 (5.0–11.0)	0.597

Median (IQR), FILS: Food Intake LEVEL scale, BMI: Body Mass Index, CPZ equivalent: Chlorpromazine-equivalent value.

**Table 2 nutrients-13-01113-t002:** Change of eating function before and after infection.

		FILS after Infection Care					Total
		2	7	8	9	10	
FILS before infection	7	3	2	0	0	0	5
	8	2	1	8	0	0	11
	9	0	0	2	2	0	4
	10	1	2	6	0	15	24
Total		6	5	16	2	15	44

FILS: Food Intake LEVEL scale; *p*-value < 0.001.

**Table 3 nutrients-13-01113-t003:** Relationship between changes in eating status and various factors.

	Maintaining Feeding Function (*n* = 30)	Declined Feeding Function (*n* = 14)	*p*-Value	Oral Intake (*n* = 38)	Parenteral Intake (*n* = 6)	*p*-Value
Age						
≥70 years, *n* (%)	16 (53.3)	10 (71.4)	0.333	21 (55.2)	5 (83.3)	0.375
Duration for treatment of infection						
≥30 days, *n* (%)	11 (36.7)	7 (50.0)	0.402	15 (39.5)	3 (50.0)	0.676
Charlson index						
Low, *n* (%)	18 (60.0)	7 (50.0)	0.412	23 (60.5)	2 (33.3)	0.060
Mild, *n* (%)	11 (37.0)	6 (42.9)		14 (36.8)	3 (50.0)	
High, *n* (%)	0 (0)	1 (0.7)		0 (0)	1 (16.7)	
Very high, *n* (%)	1 (3.0)	0 (0)		1 (2.6)	0 (0)	
CPZ equivalent (at the time of infection)						
≥1000 mg, *n* (%)	4 (13.3)	2 (14.3)	1.0	6 (15.8)	0 (0)	0.573
CPZ equivalent (during infection treatment)						
≥1000 mg, *n* (%)	5 (16.7)	0 (0)	0.16	5 (13.2)	0 (0)	1.0
Weight loss (from six months before infection)						
≥10%, *n* (%)	3 (10.0)	0 (0)	0.540	3 (7.9)	0 (0)	1.0
Weight loss (during infection treatment)						
≥10%, *n* (%)	3 (10.0)	8 (57.1)	0.002	7 (18.4)	4 (66.7)	0.027
BMI (at the time of infection)						
≥17 if less than 70, ≥17.8 if more than 70, *n* (%)	23 (76.7)	10 (71.4)	0.722	30 (78.9)	3 (50.0)	0.154
BMI (at the time of infection cure)						
≥17 if less than 70, ≥17.8 if more than 70, *n* (%)	23 (76.7)	8 (57.1)	0.186	28 (73.7)	3 (50.0)	0.339
Number of medicines (at the time of infection)						
≥6 medicines, *n* (%)	22 (73.3)	9 (64.3)	0.540	29 (76.3)	2 (33.3)	0.053
Number of medicines (during infection treatment)						
≥6 medicines, *n* (%)	15 (50.0)	6 (42.9)	0.659	18 (47.4)	3 (50.0)	1.0
Oxygen inhalation						
yes, *n* (%)	8 (26.7)	6 (42.9)	0.283	11 (28.9)	3 (50.0)	0.364
Endotracheal intubation						
yes, *n* (%)	0 (0)	2 (14.3)	0.096	2 (5.3)	0 (0)	1.0

BMI: Body Mass Index, CPZ equivalent: Chlorpromazine-equivalent value.

## Data Availability

The data presented in this study are available on request from the corresponding author.

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
