# Peer review of "COVID-19 Infection-Related Weight Loss Decreases Eating/Swallowing Function in Schizophrenic Patients"

_nutrients, 2021, doi:10.3390/nu13041113_

Round 1
Reviewer 1 Report
In this original research article, the authors examined the association between eating/swallowing function and COVID-19 infection in a population of Schizophrenic Patients. Overall, it is an interesting research idea with an important clinical meaning. However, I think that the manuscript in the present form presents major flaws. The text is sometimes dispersive and poorly focused (especially in the results/discussion sections).
Tables (especially table 1 and table 3) should be reorganized since they contain too much information and are of difficult interpretation. I recommend avoiding the term elderly along the paper since it is pejorative and reductionist. It would be better using “older people” or “older adults”.
English language needs to be improved.
Specific comments:
Page 1; line 24: please choose another word for “converted to”.
Page 1; line 33: “which started in 2019” can be removed.
Page 1; line 37: “Lancet Psychiatry” please remove it.
Page 2; line 52: In this part, it can be mentioned that COVID-19, through highly catabolic conditions, could also lead to muscle wasting (DOI: 10.1002/jcsm.12589). The loss of muscle mass and strength (i.e., sarcopenia) is not merely limited to lower limbs but is a whole body process also affecting swallowing muscles (doi.org/10.1038/s41430-020-00795-0). In fact, the concept of sarcopenic dysphagia has recently been proposed.
Page 2; line 57: Please change “many” to “a population of”
Page 2; line 72: Please rephrase.
Page 3; line 127-128: This sentence needs rephrasing as does not flow well.
Page 4; line 151-154: This paragraph needs rephrasing.
Page 7; line 221-223: Please rephrase as does not flow well.
Page 7; line 226-227: This sentence is redundant (see page 6 line 212-213)
Author Response
Comments for reviewer 1
We are grateful to reviewer 1 for the critical comments and useful suggestions that have helped us to improve our paper considerably. As indicated in the response that follow, we have taken all these comments and suggestions into account in the revised version of our paper.
Comment 1.
In this original research article, the authors examined the association between eating/swallowing function and COVID-19 infection in a population of Schizophrenic Patients. Overall, it is an interesting research idea with an important clinical meaning. However, I think that the manuscript in the present form presents major flaws. The text is sometimes dispersive and poorly focused (especially in the results/discussion sections).
Response.
Thank you for your comment.
We reviewed and revised the results/discussion sections as much as possible.
Comment 2.
Tables (especially table 1 and table 3) should be reorganized since they contain too much information and are of difficult interpretation. I recommend avoiding the term elderly along the paper since it is pejorative and reductionist. It would be better using “older people” or “older adults”.
Response.
Thank you for your comment. We have remade these tables. And we have changed corrected word from “elderly” to “older people”.
Comment 3.
Page 1; line 24: please choose another word for “converted to”.
Response.
Thank you for your useful comment. We have corrected the word from “converted to” to “transitioned to”.
Comment 4.
Page 1; line 33: “which started in 2019” can be removed.
Response.
Thank you for your useful comment. We have removed this part.
Comment 5.
Page 1; line 37: “Lancet Psychiatry” please remove it.
Response.
We appreciate your comment. We have removed the part.
Comment 6.
Page 2; line 52: In this part, it can be mentioned that COVID-19, through highly catabolic conditions, could also lead to muscle wasting (DOI: 10.1002/jcsm.12589). The loss of muscle mass and strength (i.e., sarcopenia) is not merely limited to lower limbs but is a whole body process also affecting swallowing muscles (doi.org/10.1038/s41430-020-00795-0). In fact, the concept of sarcopenic dysphagia has recently been proposed.
Response.
We appreciate your comments. We have added some sentences with reference to the presented literature as follows.
It has been pointed out that COVID-19 leads to muscle waste through a highly cata-bolic state [10]. Loss of muscle mass and strength (ie, sarcopenia) is a systemic process that affects not only the lower extremities but also the swallowing muscles [11].
Comment 5.
Page 2; line 57: Please change “many” to “a population of”
Response.
Thank you for your useful comment. We have corrected word from “many” to “a population of”
Comment 5.
Page 2; line 72: Please rephrase.
Response.
Thank you for your useful comment. We have rephrased the sentence as follows. Subjects were assigned separately according to age, ≥70 years and age, <70 years.
Comment 6.
Page 3; line 127-128: This sentence needs rephrasing as does not flow well.
Response.
Thank you for your useful comment. We have rephrased the sentence as follows. Eleven subjects were diagnosed as malnourished on the basis of BMI, and 3 subjects by the weight loss rate.
Comment 7.
Page 4; line 151-154: This paragraph needs rephrasing.
Response.
Thank you for your useful comment. We have rephrased the sentence as follows. After fully recovered from COVID-19 infection, 13 subjects were diagnosed as malnourished on the basis of BMI, and 11 subjects by the weight loss rate. The malnourished subjects who were diagnosed on the basis of weight loss rate had increased significantly compared to that before the COVID-19 infection.
Comment 8.
Page 7; line 221-223: Please rephrase as does not flow well.
Response.
Thank you for your useful comment.
We have rephrased in the text as follows.
During treatment for COVID-19, 43% of subjects in the present study experienced ≥5% weight loss, and 25% of subjects showed ≥10% weight loss. The subject of this study also suffered from the risk of malnutrition by COVID-19 infection. The subject who had 10>% weight loss showed significantly association with declined eating function and transition to parenteral feeding.
Comment 9.
Page 7; line 226-227: This sentence is redundant (see page 6 line 212-213)
Response.
Thank you for your useful comment.
We have removed this sentence.
Reviewer 2 Report
Thank you for the opportunity to review this manuscript. Overall, the manuscript is well written and interesting. However, there are some points that raise serious doubts:
1. a small sample size;
2. a large disproportion between women and men;
3. the mean age suggests that a significant proportion of your participants are elderly (it is therefore not clear whether the relationships/effects described by the authors depend on the age);
4. little is known about the severity of the symptoms and course of COVID-19 in patients;
5. the discussion largely replicates what has already been covered elsewhere in the manuscript.
Author Response
Comments for reviewer 2
We are grateful to reviewer 2 for the critical comments and useful suggestions that have helped us to improve our paper considerably. As indicated in the response that follow, we have taken all these comments and suggestions into account in the revised version of our paper.
Comment 1.
a small sample size
Response.
We appreciate your comment and we agree this point. We have described regarding to this point in the limitation section.
Comment 2. a large disproportion between women and men
Response.
We agree this point. We have added regarding to this point in the limitation section.
Comment 3. the mean age suggests that a significant proportion of your participants are elderly (it is therefore not clear whether the relationships/effects described by the authors depend on the age)
Response.
We appreciate your comment. Since we found a relationship between the severity of dysphagia and age before infection, we changed the description to focus on this part.
Comment 4. little is known about the severity of the symptoms and course of COVID-19 in patients;
Response.
We agree with this point. We have added regarding to this point in the limitation section.
Comment 5. the discussion largely replicates what has already been covered elsewhere in the manuscript.
Response.
We appreciate your comment. We reviewed and revised the discussion section as much as possible.
Round 2
Reviewer 1 Report
All my comments have been addressed. I suggest an English editing.
Author Response
We are grateful to reviewer 1 for the critical comments and useful suggestions that have helped us to improve our paper considerably.
English editing of our paper had been completed.
Reviewer 2 Report
Thank you for the opportunity to review a revised manuscript. I appreciate the authors' effort in responding to all comments. However, I still believe that there are significant limitations to this study that significantly limit its quality (a small sample size, etc.).
Author Response
We are grateful to reviewer 2 for the critical comments and useful suggestions that have helped us to improve our paper considerably.
We agree with this point. These cases that arose in a single hospital ward, was too small to allow for adequate statistical analyses.
We have added regarding to this point in the limitation section.